# Proximal Quasi-Newton for Computationally Intensive $\ell_1$-regularized $M$-estimators

**Kai Zhong** [1]    **Ian E.H. Yen** [2]    **Inderjit S. Dhillon** [2]    **Pradeep Ravikumar** [2]

[1] Institute for Computational Engineering & Sciences    [2] Department of Computer Science
University of Texas at Austin
zhongkai@ices.utexas.edu, {ianyen,inderjit,pradeepr}@cs.utexas.edu

## Abstract

We consider the class of optimization problems arising from computationally intensive $\ell_1$-regularized $M$-estimators, where the function or gradient values are very expensive to compute. A particular instance of interest is the $\ell_1$-regularized MLE for learning Conditional Random Fields (CRFs), which are a popular class of statistical models for varied structured prediction problems such as sequence labeling, alignment, and classification with label taxonomy. $\ell_1$-regularized MLEs for CRFs are particularly expensive to optimize since computing the gradient values requires an expensive inference step. In this work, we propose the use of a carefully constructed proximal quasi-Newton algorithm for such computationally intensive $M$-estimation problems, where we employ an aggressive active set selection technique. In a key contribution of the paper, we show that the proximal quasi-Newton method is provably *super-linearly convergent*, even in the absence of strong convexity, by leveraging a restricted variant of strong convexity. In our experiments, the proposed algorithm converges considerably faster than current state-of-the-art on the problems of sequence labeling and hierarchical classification.

## 1  Introduction

$\ell_1$-regularized $M$-estimators have attracted considerable interest in recent years due to their ability to fit large-scale statistical models, where the underlying model parameters are sparse. The optimization problem underlying these $\ell_1$-regularized $M$-estimators takes the form:

$$\min_{\boldsymbol{w}} f(\boldsymbol{w}) := \lambda\|\boldsymbol{w}\|_1 + \ell(\boldsymbol{w}), \tag{1}$$

where $\ell(\boldsymbol{w})$ is a convex differentiable loss function. In this paper, we are particularly interested in the case where the function or gradient values are very expensive to compute; we refer to these functions as computationally intensive functions, or **CI** functions in short. A particular case of interest are $\ell_1$-regularized MLEs for Conditional Random Fields (CRFs), where computing the gradient requires an expensive inference step.

There has been a line of recent work on computationally efficient methods for solving (1), including [2, 8, 13, 21, 23, 4]. It has now become well understood that it is key to leverage the sparsity of the optimal solution by maintaining sparse intermediate iterates [2, 5, 8]. Coordinate Descent (CD) based methods, like CDN [8], maintain the sparsity of intermediate iterates by focusing on an active set of working variables. A caveat with such methods is that, for CI functions, each coordinate update typically requires a call of inference oracle to evaluate partial derivative for single coordinate. One approach adopted in [16] to address this is using Blockwise Coordinate Descent that updates a block of variables at a time by ignoring the second-order effect, which however sacrifices the convergence guarantee. Newton-type methods have also attracted a surge of interest in recent years [5, 13], but these require computing the exact Hessian or Hessian-vector product, which is very

expensive for CI functions. This then suggests the use of quasi-Newton methods, popular instances of which include OWL-QN [23], which is adapted from $\ell_2$-regularized L-BFGS, as well as Projected Quasi-Newton (PQN) [4]. A key caveat with OWL-QN and PQN however is that they do not exploit the sparsity of the underlying solution. In this paper, we consider the class of *Proximal Quasi-Newton* (Prox-QN) methods, which we argue seem particularly well-suited to such CI functions, for the following three reasons. Firstly, it requires gradient evaluations only once in each outer iteration. Secondly, it is a second-order method, which has asymptotic superlinear convergence. Thirdly, it can employ some active-set strategy to reduce the time complexity from $O(d)$ to $O(nnz)$, where $d$ is the number of parameters and $nnz$ is the number of non-zero parameters.

While there has been some recent work on Prox-QN algorithms [2, 3], we carefully construct an implementation that is particularly suited to CI $\ell_1$-regularized $M$-estimators. We carefully maintain the sparsity of intermediate iterates, and at the same time reduce the gradient evaluation time. A key facet of our approach is our aggressive active set selection (which we also term a "shrinking strategy") to reduce the number of active variables under consideration at any iteration, and correspondingly the number of evaluations of partial gradients in each iteration. Our strategy is particularly aggressive in that it runs over multiple epochs, and in each epoch, chooses the next working set as a subset of the current working set rather than the whole set; while at the end of an epoch, allows for other variables to come in. As a result, in most iterations, our aggressive shrinking strategy only requires the evaluation of partial gradients in the current working set. Moreover, we adapt the L-BFGS update to the shrinking procedure such that the update can be conducted *without any loss of accuracy* caused by aggressive shrinking. Thirdly, we store our data in a *feature-indexed* structure to combine data sparsity as well as iterate sparsity.

[26] showed global convergence and asymptotic superlinear convergence for Prox-QN methods under the assumption that the loss function is *strongly convex*. However, this assumption is known to fail to hold in high-dimensional sampling settings, where the Hessian is typically rank-deficient, or indeed even in low-dimensional settings where there are redundant features. In a key contribution of the paper, we provide provable guarantees of asymptotic superlinear convergence for Prox-QN method, even without assuming strong-convexity, but under a restricted variant of strong convexity, termed Constant Nullspace Strong Convexity (CNSC), which is typically satisfied by standard $M$-estimators.

To summarize, our contributions are twofold. (a) We present a carefully constructed proximal quasi-Newton method for computationally intensive (CI) $\ell_1$-regularized $M$-estimators, which we empirically show to outperform many state-of-the-art methods on CRF problems. (b) We provide the first proof of asymptotic superlinear convergence for Prox-QN methods without strong convexity, but under a restricted variant of strong convexity, satisfied by typical $M$-estimators, including the $\ell_1$-regularized CRF MLEs.

## 2 Proximal Quasi-Newton Method

A proximal quasi-Newton approach to solve $M$-estimators of the form (1) proceeds by iteratively constructing a quadratic approximation of the objective function (1) to find the quasi-Newton direction, and then conducting a line search procedure to obtain the next iterate.

Given a solution estimate $\boldsymbol{w}_t$ at iteration $t$, the proximal quasi-Newton method computes a descent direction by minimizing the following regularized quadratic model,

$$\boldsymbol{d}_t = \arg \min_{\Delta} \boldsymbol{g}_t^T \Delta + \frac{1}{2} \Delta^T B_t \Delta + \lambda \|\boldsymbol{w}_t + \Delta\|_1 \tag{2}$$

where $\boldsymbol{g}_t = \boldsymbol{g}(\boldsymbol{w}_t)$ is the gradient of $\ell(\boldsymbol{w}_t)$ and $B_t$ is an approximation to the Hessian of $\ell(\boldsymbol{w})$. $B_t$ is usually formulated by the L-BFGS algorithm. This subproblem (2) can be efficiently solved by randomized coordinate descent algorithm as shown in Section 2.2.

The next iterate is obtained from the backtracking line search procedure, $\boldsymbol{w}_{t+1} = \boldsymbol{w}_t + \alpha_t \boldsymbol{d}_t$, where the step size $\alpha_t$ is tried over $\{\beta^0, \beta^1, \beta^2, ...\}$ until the Armijo rule is satisfied,

$$f(\boldsymbol{w}_t + \alpha_t \boldsymbol{d}_t) \leq f(\boldsymbol{w}_t) + \alpha_t \sigma \Delta_t,$$

where $0 < \beta < 1$, $0 < \sigma < 1$ and $\Delta_t = \boldsymbol{g}_t^T \boldsymbol{d}_t + \lambda(\|\boldsymbol{w}_t + \boldsymbol{d}_t\|_1 - \|\boldsymbol{w}_t\|_1)$.

## 2.1 BFGS update formula

$B_t$ can be efficiently updated by the gradients of the previous iterations according to the BFGS update [18],

$$B_t = B_{t-1} - \frac{B_{t-1}\boldsymbol{s}_{t-1}\boldsymbol{s}_{t-1}^T B_{t-1}}{\boldsymbol{s}_{t-1}^T B_{t-1}\boldsymbol{s}_{t-1}} + \frac{\boldsymbol{y}_{t-1}\boldsymbol{y}_{t-1}^T}{\boldsymbol{y}_{t-1}^T \boldsymbol{s}_{t-1}} \tag{3}$$

where $\boldsymbol{s}_t = \boldsymbol{w}_{t+1} - \boldsymbol{w}_t$ and $\boldsymbol{y}_t = \boldsymbol{g}_{t+1} - \boldsymbol{g}_t$
We use the compact formula for $B_t$ [18],

$$B_t = B_0 - QRQ^T = B_0 - Q\hat{Q},$$

where

$$Q := \begin{bmatrix} B_0 S_t & Y_t \end{bmatrix}, \ R := \begin{bmatrix} S_t^T B_0 S_t & L_t \\ L_t^T & -D_t \end{bmatrix}^{-1}, \hat{Q} := RQ^T$$

$$S_t = [\boldsymbol{s}_0, \boldsymbol{s}_1, ..., \boldsymbol{s}_{t-1}], \ Y_t = [\boldsymbol{y}_0, \boldsymbol{y}_1, ..., \boldsymbol{y}_{t-1}]$$

$$D_t = diag[\boldsymbol{s}_0^T \boldsymbol{y}_0, ..., \boldsymbol{s}_{t-1}^T \boldsymbol{y}_{t-1}] \text{ and } (L_t)_{i,j} = \begin{cases} \boldsymbol{s}_{i-1}^T \boldsymbol{y}_{j-1} & \text{if } i > j \\ 0 & \text{otherwise} \end{cases}$$

In practical implementation, we apply Limited-memory-BFGS. It only uses the information of the most recent $m$ gradients, so that $Q$ and $\hat{Q}$ have only size, $d \times 2m$ and $2m \times d$, respectively. $B_0$ is usually set as $\gamma_t I$ for computing $B_t$, where $\gamma_t = \boldsymbol{y}_{t-1}^T \boldsymbol{s}_{t-1} / \boldsymbol{s}_{t-1}^T \boldsymbol{s}_{t-1}$[18]. As will be discussed in Section 2.3, $Q(\hat{Q})$ is updated just on the rows(columns) corresponding to the working set, $\mathcal{A}$. The time complexity for L-BFGS update is $O(m^2|\mathcal{A}| + m^3)$.

## 2.2 Coordinate Descent for Inner Problem

Randomized coordinate descent is carefully employed to solve the inner problem (2) by Tang and Scheinberg [2]. In the update for coordinate $j$, $\boldsymbol{d} \leftarrow \boldsymbol{d} + z^* \boldsymbol{e}_j$, $z^*$ is obtained by solving the one-dimensional problem,

$$z^* = \arg\min_z \frac{1}{2}(B_t)_{jj} z^2 + ((\boldsymbol{g}_t)_j + (B_t\boldsymbol{d})_j) z + \lambda|(\boldsymbol{w}_t)_j + d_j + z|$$

This one-dimensional problem has a closed-form solution, $z^* = -c + \mathcal{S}(c - b/a, \lambda/a)$ ,where $\mathcal{S}$ is the soft-threshold function and $a = (B_t)_{jj}$, $b = (\boldsymbol{g}_t)_j + (B_t\boldsymbol{d})_j$ and $c = (\boldsymbol{w}_t)_j + d_j$. For $B_0 = \gamma_t I$, the diagonal of $B_t$ can be computed by $(B_t)_{jj} = \gamma_t - \boldsymbol{q}_j^T \hat{\boldsymbol{q}}_j$, where $\boldsymbol{q}_j^T$ is the j-th row of $Q$ and $\hat{\boldsymbol{q}}_j$ is the j-th column of $\hat{Q}$. And the second term in $b$, $(B_t\boldsymbol{d})_j$ can be computed by,

$$(B_t\boldsymbol{d})_j = \gamma_t d_j - \boldsymbol{q}_j^T \hat{Q}\boldsymbol{d} = \gamma_t d_j - \boldsymbol{q}_j^T \hat{\boldsymbol{d}},$$

where $\hat{\boldsymbol{d}} := \hat{Q}\boldsymbol{d}$. Since $\hat{\boldsymbol{d}}$ has only $2m$ dimension, it is fast to update $(B_t\boldsymbol{d})_j$ by $\boldsymbol{q}_j$ and $\hat{\boldsymbol{d}}$. In each inner iteration, only $d_j$ is updated, so we have the fast update of $\hat{\boldsymbol{d}}$, $\hat{\boldsymbol{d}} \leftarrow \hat{\boldsymbol{d}} + \hat{\boldsymbol{q}}_j z^*$.

Since we only update the coordinates in the working set, the above algorithm has only computation complexity $O(m|\mathcal{A}| \times inner\_iter)$, where $inner\_iter$ is the number of iterations used for solving the inner problem.

## 2.3 Implementation

In this section, we discuss several key implementation details used in our algorithm to speed up the optimization.

**Shrinking Strategy**
In each iteration, we select an active or working subset $\mathcal{A}$ of the set of all variables: only the variables in this set are updated in the current iteration. The complementary set, also called the fixed set, has only values of zero and is not updated. The use of such a shrinking strategy reduces the overall complexity from $O(d)$ to $O(|\mathcal{A}|)$. Specifically, we (a) update the gradients just on the working set, (b) update $Q$ ($\hat{Q}$) just on the rows(columns) corresponding to the working set, and (c) compute the latest entries in $D_t$, $\gamma_t$, $L_t$ and $S_t^T S_t$ by just using the corresponding working set rather than the whole set.

The key facet of our "shrinking strategy" however is in aggressively shrinking the active set: at the next iteration, we set the active set to be a subset of the previous active set, so that $\mathcal{A}_t \subset \mathcal{A}_{t-1}$. Such an aggressive shrinking strategy however is not guaranteed to only weed out irrelevant variables. Accordingly, we proceed in epochs. In each epoch, we progressively shrink the active set as above, till the iterations seem to converge. At that time, we then allow for all the "shrunk" variables to come back and start a new epoch. Such a strategy was also called an $\epsilon$-cooling strategy by Fan et al. [14], where the shrinking stopping criterion is loose at the beginning, and progressively becomes more strict each time all the variables are brought back. For L-BFGS update, when a new epoch starts, the memory of L-BFGS is cleaned to prevent any loss of accuracy.

Because at the first iteration of each new epoch, the entire gradient over all coordinates is evaluated, the computation time for those iterations accounts for a significant portion of the total time complexity. Fortunately, our experiments show that the number of epochs is typically between 3-5.

**Inexact inner problem solution**
Like many other proximal methods, e.g. GLMNET and QUIC, we solve the inner problem inexactly. This reduces the time complexity of the inner problem dramatically. The amount of inexactness is based on a heuristic method which aims to balance the computation time of the inner problem in each outer iteration. The computation time of the inner problem is determined by the number of inner iterations and the size of working set. Thus, we let the number of inner iterations, $inner\_iter = \min\{max\_inner, \lfloor d/|\mathcal{A}| \rfloor\}$, where $max\_inner = 10$ in our experiment.

**Data Structure for both model sparsity and data sparsity**
In our implementation we take two sparsity patterns into consideration: (a) model sparsity, which accounts for the fact that most parameters are equal to zero in the optimal solution; and (b) data sparsity, wherein most feature values of any particular instance are zeros. We use a *feature-indexed* data structure to take advantage of both sparsity patterns. Computations involving data will be time-consuming if we compute over all the instances including those that are zero. So we leverage the sparsity of data in our experiment by using vectors of pairs, whose members are the index and its value. Traditionally, each vector represents an instance and the indices in its pairs are the feature indices. However, in our implementation, to take both model sparsity and data sparsity into account, we use an inverted data structure, where each vector represents one feature (*feature-indexed*) and the indices in its pairs are the instance indices. This data structure facilitates the computation of the gradient for a particular feature, which involves only the instances related to this feature.

We summarize these steps in the algorithm below. And a detailed algorithm is in Appendix 2.

---
**Algorithm 1** Proximal Quasi-Newton Algorithm (Prox-QN)

---
**Input:** Dataset $\{\boldsymbol{x}^{(i)}, \boldsymbol{y}^{(i)}\}_{i=1,2,\ldots,N}$, termination criterion $\epsilon$, $\lambda$ and L-BFGS memory size $m$.
**Output:** $\boldsymbol{w}^*$ converging to arg $\min_{\boldsymbol{w}} f(\boldsymbol{w})$.
  1: Initialize $\boldsymbol{w} \leftarrow \boldsymbol{0}$, $\boldsymbol{g} \leftarrow \partial \ell(\boldsymbol{w})/\partial \boldsymbol{w}$, working set $\mathcal{A} \leftarrow \{1, 2, \ldots d\}$, and $S, Y, Q, \hat{Q} \leftarrow \phi$.
  2: **while** termination criterion is not satisfied or working set doesn't contain all the variables **do**
  3:      Shrink working set.
  4:      **if** Shrinking stopping criterion is satisfied **then**
  5:          Take all the shrunken variables back to working set and clean the memory of L-BFGS.
  6:          Update Shrinking stopping criterion and continue.
  7:      **end if**
  8:      Solve inner problem (2) over working set and obtain the new direction $\boldsymbol{d}$.
  9:      Conduct line search based on Armijo rule and obtain new iterate $\boldsymbol{w}$.
 10:      Update $\boldsymbol{g}$, s, $\boldsymbol{y}$, $S, Y, Q, \hat{Q}$ and related matrices over working set.
 11: **end while**

---

## 3   Convergence Analysis

In this section, we analyze the convergence behavior of proximal quasi-Newton method in the superlinear convergence phase, where the unit step size is chosen. To simplify the analysis, in this section, we assume the inner problem is solved exactly and no shrinking strategy is employed. We also provide the global convergence proof for Prox-QN method with shrinking strategy in Appendix 1.5. In current literature, the analysis of proximal Newton-type methods relies on the assumption of

*strongly convex* objective function to prove superlinear convergence [3]; otherwise, only sublinear rate can be proved [25]. However, our objective (1) is not strongly convex when the dimension is very large or there are redundant features. In particular, the Hessian matrix $H(\boldsymbol{w})$ of the smooth function $\ell(\boldsymbol{w})$ is not positive-definite. We thus leverage a recently introduced restricted variant of strong convexity, termed Constant Nullspace Strong Convexity (CNSC) in [1]. There the authors analyzed the behavior of proximal gradient and proximal Newton methods under such a condition. The proximal *quasi-Newton* procedure in this paper however requires a subtler analysis, but in a key contribution of the paper, we are nonetheless able to show asymptotic superlinear convergence of the Prox-QN method under this restricted variant of strong convexity.

**Definition 1** (Constant Nullspace Strong Convexity (CNSC)). *A composite function* (1) *is said to have Constant Nullspace Strong Convexity restricted to space $\mathcal{T}$ (CNSC-$\mathcal{T}$) if there is a constant vector space $\mathcal{T}$ s.t. $\ell(\boldsymbol{w})$ depends only on $\mathbf{proj}_{\mathcal{T}}(\boldsymbol{w})$, i.e. $\ell(\boldsymbol{w}) = \ell(\mathbf{proj}_{\mathcal{T}}(\boldsymbol{w}))$, and its Hessian satisfies*

$$m\|\boldsymbol{v}\|^2 \leq \boldsymbol{v}^T H(\boldsymbol{w})\boldsymbol{v} \leq M\|\boldsymbol{v}\|^2, \quad \forall \boldsymbol{v} \in \mathcal{T}, \forall \boldsymbol{w} \in \mathbb{R}^d \tag{4}$$

*for some $M \geq m > 0$, and*

$$H(\boldsymbol{w})\boldsymbol{v} = \boldsymbol{0}, \quad \forall \boldsymbol{v} \in \mathcal{T}^{\perp}, \forall \boldsymbol{w} \in \mathbb{R}^d, \tag{5}$$

*where $\mathbf{proj}_{\mathcal{T}}(\boldsymbol{w})$ is the projection of $\boldsymbol{w}$ onto $\mathcal{T}$ and $\mathcal{T}^{\perp}$ is the complementary space orthogonal to $\mathcal{T}$.*

This condition can be seen to be an algebraic condition that is satisfied by typical $M$-estimators considered in high-dimensional settings. In this paper, we will abuse the use of CNSC-$\mathcal{T}$ for symmetric matrices. We say a symmetric matrix $H$ satisfies CNSC-$\mathcal{T}$ condition if $H$ satisfies (4) and (5). In the following theorems, we will denote the orthogonal basis of $\mathcal{T}$ as $U \in \mathbb{R}^{d \times \hat{d}}$, where $\hat{d} \leq d$ is the dimensionality of $\mathcal{T}$ space and $U^T U = I$. Then the projection to $\mathcal{T}$ space can be written as $\mathbf{proj}_{\mathcal{T}}(\boldsymbol{w}) = UU^T\boldsymbol{w}$.

**Theorem 1** (Asymptotic Superlinear Convergence). *Assume $\nabla^2 \ell(\boldsymbol{w})$ and $\nabla \ell(\boldsymbol{w})$ are Lipschitz continuous. Let $B_t$ be the matrices generated by BFGS update (3). Then if $\ell(\boldsymbol{w})$ and $B_t$ satisfy CNSC-$\mathcal{T}$ condition, the proximal quasi-Newton method has q-superlinear convergence:*

$$\|\boldsymbol{z}_{t+1} - \boldsymbol{z}^*\| \leq o\left(\|\boldsymbol{z}_t - \boldsymbol{z}^*\|\right),$$

*where $\boldsymbol{z}_t = U^T\boldsymbol{w}_t$, $\boldsymbol{z}^* = U^T\boldsymbol{w}^*$ and $\boldsymbol{w}^*$ is an optimal solution of* (1).

The proof is given in Appendix 1.4. We prove it by exploiting the CNSC-$\mathcal{T}$ property. First, we re-build our problem and algorithm on the reduced space $\mathcal{Z} = \{\boldsymbol{z} \in \mathbb{R}^{\hat{d}} | \boldsymbol{z} = U^T\boldsymbol{w}\}$, where the strong-convexity property holds. Then we prove the asymptotic superlinear convergence on $\mathcal{Z}$ following Theorem 3.7 in [26].

**Theorem 2.** *For Lipschitz continuous $\ell(\boldsymbol{w})$, the sequence $\{\boldsymbol{w}_t\}$ produced by the proximal quasi-Newton Method in the super-linear convergence phase has*

$$f(\boldsymbol{w}_t) - f(\boldsymbol{w}^*) \leq L\|\boldsymbol{z}_t - \boldsymbol{z}^*\|, \tag{6}$$

*where $L = L_\ell + \lambda\sqrt{d}$, $L_\ell$ is the Lipschitz constant of $\ell(\boldsymbol{w})$, $\boldsymbol{z}_t = U^T\boldsymbol{w}_t$ and $\boldsymbol{z}^* = U^T\boldsymbol{w}^*$.*

The proof is also in Appendix 1.4. It is proved by showing that both the smooth part and the non-differentiable part satisfy the modified Lipschitz continuity.

## 4  Application to Conditional Random Fields with $\ell_1$ Penalty

In CRF problems, we are interested in learning a conditional distribution of labels $\boldsymbol{y} \in \mathcal{Y}$ given observation $\boldsymbol{x} \in \mathcal{X}$, where $\boldsymbol{y}$ has application-dependent structure such as sequence, tree, or table in which label assignments have inter-dependency. The distribution is of the form

$$P_{\boldsymbol{w}}(\boldsymbol{y}|\boldsymbol{x}) = \frac{1}{Z_{\boldsymbol{w}}(\boldsymbol{x})} \exp\left\{\sum_{k=1}^d w_k f_k(\boldsymbol{y}, \boldsymbol{x})\right\},$$

where $f_k$ is the feature functions, $w_k$ is the associated weight, $d$ is the number of feature functions and $Z_{\boldsymbol{w}}(\boldsymbol{x})$ is the partition function. Given a training data set $\{(\boldsymbol{x}_i, \boldsymbol{y}_i)\}_{i=1}^N$, our goal is to find the optimal weights $\boldsymbol{w}$ such that the following $\ell_1$-regularized negative log-likelihood is minimized.

$$\min_{\boldsymbol{w}} f(\boldsymbol{w}) = \lambda\|\boldsymbol{w}\|_1 - \sum_{i=1}^N \log P_{\boldsymbol{w}}(\boldsymbol{y}^{(i)}|\boldsymbol{x}^{(i)}) \tag{7}$$

Since $|\mathcal{Y}|$, the number of possible values $\boldsymbol{y}$ takes, can be exponentially large, the evaluation of $\ell(\boldsymbol{w})$ and the gradient $\nabla\ell(\boldsymbol{w})$ needs application-dependent oracles to conduct the summation over $\mathcal{Y}$. For example, in *sequence labeling problem*, a dynamic programming oracle, *forward-backward* algorithm, is usually employed to compute $\nabla\ell(\boldsymbol{w})$. Such an oracle can be very expensive. In Prox-QN algorithm for sequence labeling problem, the *forward-backward* algorithm takes $O(|Y|^2 NT \times exp)$ time, where $exp$ is the time for the expensive exponential computation, $T$ is the sequence length and $Y$ is the possible label set for a symbol in the sequence. Then given the obtained oracle, the evaluation of the partial gradients over the working set $\mathcal{A}$ has time complexity, $O(D_{nnz}|\mathcal{A}|T)$, where $D_{nnz}$ is the average number of instances related to a feature. Thus when $O(|Y|^2 NT \times exp + D_{nnz}|\mathcal{A}|T) > O(m^3 + m^2|\mathcal{A}|)$, the gradients evaluation time will dominate.

The following theorem gives that the $\ell_1$-regularized CRF MLEs satisfy the CNSC-$\mathcal{T}$ condition.

**Theorem 3.** *With $\ell_1$ penalty, the CRF loss function, $\ell(\boldsymbol{w}) = -\sum_{i=1}^{N} \log P_{\boldsymbol{w}}(\boldsymbol{y}^{(i)}|\boldsymbol{x}^{(i)})$, satisfies the CNSC-$\mathcal{T}$ condition with $\mathcal{T} = \mathcal{N}^{\perp}$, where $\mathcal{N} = \{\boldsymbol{v} \in \mathbb{R}^d | \Phi^T \boldsymbol{v} = 0\}$ is a constant subspace of $\mathbb{R}^d$ and $\Phi \in \mathbb{R}^{d \times (N|\mathcal{Y}|)}$ is defined as below,*

$$\Phi_{kn} = f_k(\boldsymbol{y}_l, \boldsymbol{x}^{(i)}) - E\left[f_k(\boldsymbol{y}, \boldsymbol{x}^{(i)})\right]$$

*where $n = (i-1)|\mathcal{Y}| + l$, $l = 1, 2, ...|\mathcal{Y}|$ and $E$ is the expectation over the conditional probability $P_{\boldsymbol{w}}(\boldsymbol{y}|\boldsymbol{x}^{(i)})$.*

According to the definition of CNSC-$\mathcal{T}$ condition, the $\ell_1$-regularized CRF MLEs don't satisfy the classical strong-convexity condition when $\mathcal{N}$ has non-zero members, which happens in the following two cases: (i) the exponential representation is not minimal [27], i.e. for any instance $i$ there exist a non-zero vector $\boldsymbol{a}$ and a constant $b_i$ such that $\langle \boldsymbol{a}, \phi(\boldsymbol{y}, \boldsymbol{x}^{(i)}) \rangle = b_i$, where $\phi(\boldsymbol{y}, \boldsymbol{x}) = [f_1(\boldsymbol{y}, \boldsymbol{x}^{(i)}), f_2(\boldsymbol{y}, \boldsymbol{x}^{(i)}), ..., f_d(\boldsymbol{y}, \boldsymbol{x}^{(i)})]^T$; (ii) $d > N|\mathcal{Y}|$, i.e., the number of feature functions is very large. The first case holds in many problems, like the sequence labeling and hierarchical classification discussed in Section 6, and the second case will hold in high-dimensional problems.

## 5 Related Methods

There have been several methods proposed for solving $\ell_1$-regularized $M$-estimators of the form in (7). In this section, we will discuss these in relation to our method.

**Orthant-Wise Limited-memory Quasi-Newton** (OWL-QN) introduced by Andrew and Gao [23] extends L-BFGS to $\ell_1$-regularized problems. In each iteration, OWL-QN computes a generalized gradient called *pseudo-gradient* to determine the orthant and the search direction, then does a line search and a projection of the new iterate back to the orthant. Due to its fast convergence, it is widely implemented by many software packages, such as CRF++, CRFsuite and Wapiti. But OWL-QN does not take advantage of the model sparsity in the optimization procedure, and moreover Yu et al. [22] have raised issues with its convergence proof.

**Stochastic Gradient Descent** (SGD) uses the gradient of a single sample as the search direction at each iteration. Thus, the computation for each iteration is very fast, which leads to fast convergence at the beginning. However, the convergence becomes slower than the second-order method when the iterate is close to the optimal solution. Recently, an $\ell_1$-regularized SGD algorithm proposed by Tsuruoka et al.[21] is claimed to have faster convergence than OWL-QN. It incorporates $\ell_1$-regularization by using a cumulative $\ell_1$ penalty, which is close to the $\ell_1$ penalty received by the parameter if it had been updated by the true gradient. Tsuruoka et al. do consider data sparsity, i.e. for each instance, only the parameters related to the current instance are updated. But they too do not take the model sparsity into account.

**Coordinate Descent** (CD) and **Blockwise Coordinate Descent** (BCD) are popular methods for $\ell_1$-regularized problem. In each coordinate descent iteration, it solves an one-dimensional quadratic approximation of the objective function, which has a closed-form solution. It requires the second partial derivative with respect to the coordinate. But as discussed by Sokolovska et al., the exact second derivative in CRF problem is intractable. So they instead use an approximation of the second derivative, which can be computed efficiently by the same inference oracle queried for the gradient evaluation. However, pure CD is very expensive because it requires to call the inference oracle for the instances related to the current coordinate in each coordinate update. BCD alleviates this problem by grouping the parameters with the same $\boldsymbol{x}$ feature into a block. Then each block update only

needs to call the inference oracle once for the instances related to the current $\boldsymbol{x}$ feature. However, it cannot alleviate the large number of inference oracle calls unless the data is very sparse such that every instance appears only in very few blocks.

**Proximal Newton method** has proven successful on problems of $\ell_1$-regularized logistic regression [13] and Sparse Invariance Covariance Estimation [5], where the Hessian-vector product can be cheaply re-evaluated for each update of coordinate. However, the Hessian-vector product for CI function like CRF requires the query of the inference oracle no matter how many coordinates are updated at a time [17], which then makes the coordinate update on quadratic approximation as expensive as coordinate update in the original problem. Our proximal quasi-Newton method avoids such problem by replacing Hessian with a low-rank matrix from BFGS update.

## 6 Numerical Experiments

We compare our approach, Prox-QN, with four other methods, Proximal Gradient (Prox-GD), OWL-QN [23], SGD [21] and BCD [16]. For OWL-QN, we directly use the OWL-QN optimizer developed by Andrew et al.[1], where we set the memory size as $m = 10$, which is the same as that in Prox-QN. For SGD, we implement the algorithm proposed by Tsuruoka et al. [21], and use cumulative $\ell_1$ penalty with learning rate $\eta_k = \eta_0/(1 + k/N)$, where k is the SGD iteration and $N$ is the number of samples. For BCD, we follow Sokolovska et al. [16] but with three modifications. First, we add a line search procedure in each block update since we found it is required for convergence. Secondly, we apply shrinking strategy as discussed in Section 2.3. Thirdly, when the second derivative for some coordinate is less than $10^{-10}$, we set it to be $10^{-10}$ because otherwise the lack of $\ell_2$-regularization in our problem setting will lead to a very large new iterate.

We evaluate the performance of Prox-QN method on two problems, sequence labeling and hierarchical classification. In particular, we plot the relative objective difference $(f(\boldsymbol{w}_t) - f(\boldsymbol{w}^*))/f(\boldsymbol{w}^*)$ and the number of non-zero parameters (on a log scale) against time in seconds. More experiment results, for example, the testing accuracy and the performance for different $\lambda$'s, are in Appendix 5. All the experiments are executed on 2.8GHz Intel Xeon E5-2680 v2 Ivy Bridge processor with 1/4TB memory and Linux OS.

### 6.1 Sequence Labeling

In sequence labeling problems, each instance $(\boldsymbol{x}, \boldsymbol{y}) = \{(\boldsymbol{x}_t, y_t)\}_{t=1,2...,T}$ is a sequence of $T$ pairs of observations and the corresponding labels. Here we consider the optical character recognition (OCR) problem, which aims to recognize the handwriting words. The dataset [2] was preprocessed by Taskar et al. [19] and was originally collected by Kassel [20], and contains 6877 words (instances). We randomly divide the dataset into two part: training part with 6216 words and testing part with 661 words. The character label set $Y$ consists of 26 English letters and the observations are characters which are represented by images of 16 by 8 binary pixels as shown in Figure 1(a). We use degree 2 pixels as the raw features, which means all pixel pairs are considered. Therefore, the number of raw features is $J = 128 \times 127/2 + 128 + 1$, including a bias. For degree 2 features, $x_{tj} = 1$ only when both pixels are 1 and otherwise $x_{tj} = 0$, where $x_{tj}$ is the j-th raw feature of $\boldsymbol{x}_i$. For the feature functions, we use unigram feature functions $\mathbf{1}(y_t = y, x_{tj} = 1)$ and bigram feature functions $\mathbf{1}(y_t = y, y_{t+1} = y')$ with their associated weights, $\Theta_{y,j}$ and $\Lambda_{y,y'}$, respectively. So $\boldsymbol{w} = \{\Theta, \Lambda\}$ for $\Theta \in \mathbb{R}^{|Y| \times J}$ and $\Lambda \in \mathbb{R}^{|Y| \times |Y|}$ and the total number of parameters, $d = |Y|^2 + |Y| \times J = 215,358$. Using the above feature functions, the potential function can be specified as, $\tilde{P}_{\boldsymbol{w}}(\boldsymbol{y}, \boldsymbol{x}) = \exp\left\{\langle\Lambda, \sum_{t=1}^{T}(e_{y_t}\boldsymbol{x}_t^T)\rangle + \langle\Theta, \sum_{t=1}^{T-1}(e_{y_t}e_{y_{t+1}}^T)\rangle\right\}$, where $\langle\cdot,\cdot\rangle$ is the sum of element-wise product and $\boldsymbol{e}_y \in \mathbb{R}^{|Y|}$ is an unit vector with 1 at y-th entry and 0 at other entries. The gradient and the inference oracle are given in Appendix 4.1.

In our experiment, $\lambda$ is set as 100, which leads to a relative high testing accuracy and an optimal solution with a relative small number of non-zero parameters (see Appendix 5.2). The learning rate $\eta_0$ for SGD is tuned to be $2 \times 10^{-4}$ for best performance. In BCD, the unigram parameters are grouped into $J$ blocks according to the $\boldsymbol{x}$ features while the bigram parameters are grouped into one block. Our proximal quasi-Newton method can be seen to be much faster than the other methods.

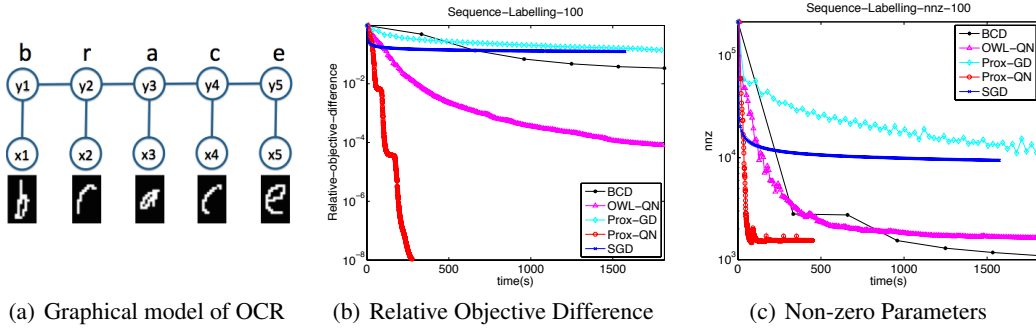

(a) Graphical model of OCR     (b) Relative Objective Difference     (c) Non-zero Parameters

Figure 1: Sequence Labeling Problem

## 6.2 Hierarchical Classification

In hierarchical classification problems, we have a label taxonomy, where the classes are grouped into a tree as shown in Figure 2(a). Here $y \in \mathcal{Y}$ is one of the leaf nodes. If we have totally $K$ classes (number of nodes) and $J$ raw features, then the number of parameters is $d = K \times J$. Let $W \in \mathbb{R}^{K \times J}$ denote the weights. The feature function corresponding to $W_{k,j}$ is $f_{k,j}(y, \boldsymbol{x}) = \mathbf{1}[k \in \text{Path}(y)]x_j$, where $k \in$ Path($y$) means class $k$ is an ancestor of $y$ or $y$ itself. The potential function is $\tilde{P}_W(y, \boldsymbol{x}) = \exp \left\{ \sum_{k \in \text{Path}(y)} \boldsymbol{w}_k^T \boldsymbol{x} \right\}$ where $\boldsymbol{w}_k^T$ is the weight vector of k-th class, i.e. the k-th row of $W$. The gradient and the inference oracle are given in Appendix 4.2.

The dataset comes from Task1 of the dry-run dataset of LSHTC1[3]. It has 4,463 samples, each with $J$=51,033 raw features. The hierarchical tree has 2,388 classes which includes 1,139 leaf labels. Thus, the number of the parameters $d$ =121,866,804. The feature values are scaled by svm-scale program in the LIBSVM package. We set $\lambda = 1$ to achieve a relative high testing accuracy and high sparsity of the optimal solution. The SGD initial learning rate is tuned to be $\eta_0 = 10$ for best performance. In BCD, parameters are grouped into $J$ blocks according to the raw features.

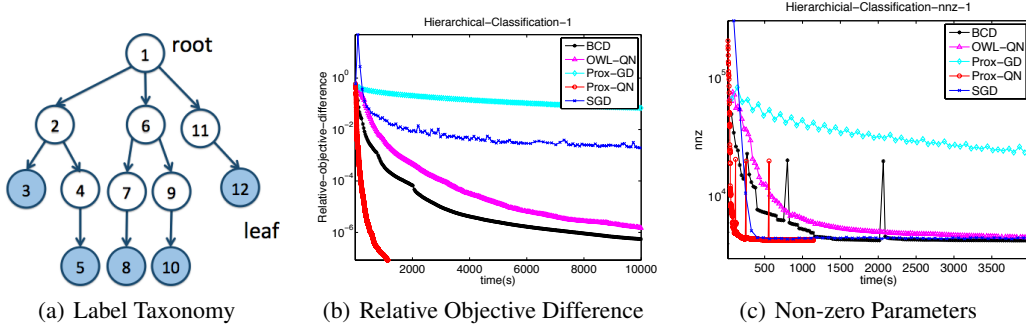

(a) Label Taxonomy     (b) Relative Objective Difference     (c) Non-zero Parameters

Figure 2: Hierarchical Classification Problem

As both Figure 1(b),1(c) and Figure 2(b),2(c) show, Prox-QN achieves much faster convergence and moreover obtains a sparse model in much less time.

## Acknowledgement

This research was supported by NSF grants CCF-1320746 and CCF-1117055. P.R. acknowledges the support of ARO via W911NF-12-1-0390 and NSF via IIS-1149803, IIS-1320894, IIS-1447574, and DMS-1264033. K.Z. acknowledges the support of the National Initiative for Modeling and Simulation fellowship

## Footnotes

[1]http://research.microsoft.com/en-us/downloads/b1eb1016-1738-4bd5-83a9-370c9d498a03/

[2]http://www.seas.upenn.edu/ taskar/ocr/

[3]http://lshtc.iit.demokritos.gr/node/1

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
