[Supplementary Material · Appendix.pdf]

# Appendix: Proximal Quasi-Newton for Computationally Intensive $\ell_1$-regularized $M$-estimators

**Kai Zhong** [1]     **Ian E.H. Yen** [2]     **Inderjit S. Dhillon** [2]     **Pradeep Ravikumar** [2]
[1] Institute for Computational Engineering & Sciences    [2] Department of Computer Science
University of Texas at Austin
zhongkai@ices.utexas.edu, {ianyen,inderjit,pradeepr}@cs.utexas.edu

The objective function is,

$$\min_{\boldsymbol{w}} f(\boldsymbol{w}) := \lambda \|\boldsymbol{w}\|_1 + \ell(\boldsymbol{w}), \tag{1}$$

## 1 Convergence Proof

**Definition 1** (Constant Nullspace Strong Convexity). *An composite function* (1) *is said to have Constant Nullspace Strong Convexity (CNSC) restricted to space $\mathcal{T}$ (CNSC-$\mathcal{T}$) iff there is a constant vector space $\mathcal{T}$ s.t. $\ell(\boldsymbol{w})$ depends only on $\boldsymbol{z} = \mathbf{proj}_{\mathcal{T}}(\boldsymbol{w})$, i.e. $\ell(\boldsymbol{w}) = \ell(\boldsymbol{z})$, and its Hessian satisfies*

$$m\|\boldsymbol{v}\|^2 \le \boldsymbol{v}^T H(\boldsymbol{w})\boldsymbol{v} \le M\|\boldsymbol{v}\|^2, \quad \forall \boldsymbol{v} \in \mathcal{T}, \forall \boldsymbol{w} \in \mathbb{R}^d \tag{2}$$

*for some $M \ge m > 0$, and*

$$H(\boldsymbol{w})\boldsymbol{v} = \mathbf{0}, \quad \forall \boldsymbol{v} \in \mathcal{T}^{\perp}, \forall \boldsymbol{w} \in \mathbb{R}^d, \tag{3}$$

*where $\mathcal{T}^{\perp}$ is the complementary space orthogonal to $\mathcal{T}$.*

To exploit the CNSC-$\mathcal{T}$ property, we first re-build our problem and algorithm on the reduced space $\mathcal{Z} = \{\boldsymbol{z} \in \mathbb{R}^{\hat{d}} | \boldsymbol{z} = U^T \boldsymbol{w}\}$, where the strong-convexity property holds. Then we prove the asymptotic super-linear convergence on $\mathcal{Z}$ under the condition that the inner problem is solved exactly and no shrinking strategy is not applied. Finally we prove the objective (1) is bounded by the difference between current iterate and the optimal solution. In Section 1.5, we provide the global convergence proof when the shrinking strategy is applied.

### 1.1 Representing the problem in a reduced and compact space

**Properties of CNSC-$\mathcal{T}$ condition**
For $\ell(\boldsymbol{w})$ satisfying CNSC-$\mathcal{T}$ condition, we have $\ell(\boldsymbol{w}) = \ell(\mathbf{proj}_{\mathcal{T}}(\boldsymbol{w}))$. Define $\boldsymbol{g}$ to be the gradient of $\ell(\boldsymbol{w})$ and $H$ to be the Hessian of $\ell(\boldsymbol{w})$. As both $\boldsymbol{g}$ and $H$ are in the $\mathcal{T}$ space, we have $\boldsymbol{g}(\boldsymbol{w}) = UU^T \boldsymbol{g}(\mathbf{proj}_{\mathcal{T}}(\boldsymbol{w})) = \boldsymbol{g}(\mathbf{proj}_{\mathcal{T}}(\boldsymbol{w}))$ and $H(\boldsymbol{w}) = UU^T H(\mathbf{proj}_{\mathcal{T}}(\boldsymbol{w}))UU^T = H(\mathbf{proj}_{\mathcal{T}}(\boldsymbol{w}))$.

**Objective formulation in the reduced space**
Define $\hat{\ell}(\boldsymbol{z}) = \ell(U\boldsymbol{z})$. Then if $\boldsymbol{z} = U^T \boldsymbol{w}$, we have $\hat{\ell}(\boldsymbol{z}) = \ell(\boldsymbol{w})$, $\hat{\boldsymbol{g}}(\boldsymbol{z}) = U^T \boldsymbol{g}(\boldsymbol{w})$ and $\hat{H}(\boldsymbol{z}) = U^T H(\boldsymbol{w})U$, where $\hat{\boldsymbol{g}}(\boldsymbol{z})$ and $\hat{H}(\boldsymbol{z})$ are the gradient and Hessian of $\hat{\ell}(\boldsymbol{z})$ respectively. Now $\hat{H}$ is positive definite with minimal eigenvalue $m$. The objective (1) can be re-formulated in the reduced space by

$$\min_{\boldsymbol{z}} \hat{f}(\boldsymbol{z}) = h(\boldsymbol{z}) + \hat{\ell}(\boldsymbol{z}), \tag{4}$$

where

$$h(\boldsymbol{z}) = \min_{U^T \boldsymbol{w} = \boldsymbol{z}} \lambda \|\boldsymbol{w}\|_1$$

We now prove that $h(\boldsymbol{z})$ is a convex function, i.e.,

$$ch(\boldsymbol{z}_1) + (1-c)h(\boldsymbol{z}_2) \geq h(c\boldsymbol{z}_1 + (1-c)\boldsymbol{z}_2)$$

for any $0 \leq c \leq 1$, $\boldsymbol{z}_1$ and $\boldsymbol{z}_2$.

*Proof.* Let

$$\boldsymbol{w}_1 = \operatorname*{argmin}_{U^T \boldsymbol{w} = \boldsymbol{z}_1} \lambda \|w\|_1 \text{ and } \boldsymbol{w}_2 = \operatorname*{argmin}_{U^T \boldsymbol{w} = \boldsymbol{z}_2} \lambda \|w\|_1$$

Then,

$$
\begin{aligned}
ch(\boldsymbol{z}_1) + (1-c)h(\boldsymbol{z}_2) &= \lambda(c\|\boldsymbol{w}_1\|_1 + (1-c)\|\boldsymbol{w}_2\|_1) \\
&\geq \lambda(\|c\boldsymbol{w}_1 + (1-c)\boldsymbol{w}_2\|_1) \\
&\geq h(U^T(c\boldsymbol{w}_1 + (1-c)\boldsymbol{w}_2)) \\
&= h(c\boldsymbol{z}_1 + (1-c)\boldsymbol{z}_2)
\end{aligned}
$$

$\square$

The optimal solution $\boldsymbol{z}^*$ of (4) has the following relationship with the optimal solution $\boldsymbol{w}^*$ of (1),

$$\boldsymbol{w}^* = \operatorname*{argmin}_{U^T \boldsymbol{w} = \boldsymbol{z}^*} \lambda \|\boldsymbol{w}\|_1 \text{ and } \boldsymbol{z}^* = U^T \boldsymbol{w}^* \tag{5}$$

**Lipschitz continuity in the reduced space**
Throughout the paper, we assume the Hessian of $\ell(\boldsymbol{w})$ has Lipschitz continuity with constant $L_H$. According to the Lipschitz continuity,

$$\|H(\boldsymbol{w}_2)(\boldsymbol{w}_1 - \boldsymbol{w}_2) - (\boldsymbol{g}(\boldsymbol{w}_1) - \boldsymbol{g}(\boldsymbol{w}_2))\| \leq \frac{L_H}{2}\|\boldsymbol{w}_1 - \boldsymbol{w}_2\|^2$$

In the corresponding reduced space, the Lipschitz continuity also holds with the same constant .

$$\|\hat{H}(\boldsymbol{z}_2)(\boldsymbol{z}_1 - \boldsymbol{z}_2) - (\hat{\boldsymbol{g}}(\boldsymbol{z}_1) - \hat{\boldsymbol{g}}(\boldsymbol{z}_2))\| \leq \frac{L_H}{2}\|\boldsymbol{z}_1 - \boldsymbol{z}_2\|^2 \tag{6}$$

**BFGS update formula in the reduced space**
If $B_0$ is in the $\mathcal{T}$ space, $B_t$ is also in the $\mathcal{T}$ space. This can be shown by re-formulating the BFGS update and mathematical induction,

$$B_t = U\hat{B}_{t-1}U^T - \frac{U\hat{B}_{t-1}U^T s_{t-1}s_{t-1}^T U\hat{B}_{t-1}U^T}{s_{t-1}^T U\hat{B}_{t-1}U^T s_{t-1}} + \frac{UU^T y_{t-1}y_{t-1}^T UU^T}{y_{t-1}^T UU^T s_{t-1}} \tag{7}$$

Thus

$$\hat{B}_t = \hat{B}_{t-1} - \frac{\hat{B}_{t-1}\hat{s}_{t-1}\hat{s}_{t-1}^T \hat{B}_{t-1}}{\hat{s}_{t-1}^T \hat{B}_{t-1}\hat{s}_{t-1}} + \frac{\hat{y}_{t-1}\hat{y}_{t-1}^T}{\hat{y}_{t-1}^T \hat{s}_{t-1}} \tag{8}$$

where $\hat{s} = U^T s$, $\hat{y} = U^T y$ and $U\hat{B}_t U^T = B_t$. It can be proved that $\hat{B}_t$ generated in (8) is positive definite provided $\hat{y}^T \hat{s} > 0$ [2]. If we additionally assume $m\|\boldsymbol{z}\|^2 \leq \boldsymbol{z}^T \hat{B}_t \boldsymbol{z} \leq M\|\boldsymbol{z}\|^2$ for any $\boldsymbol{z} \in \mathbb{R}^{\hat{d}}$, then $B_t$ satisfies the CNSC-$\mathcal{T}$ condition.

**Iterate in the reduced space**
The potential new iterate $\boldsymbol{w}^+$ is

$$\boldsymbol{w}^+ = \operatorname*{argmin}_{\boldsymbol{v}} \lambda \|\boldsymbol{v}\|_1 + \frac{1}{2}(\boldsymbol{v} - \boldsymbol{w}_t)^T B_t(\boldsymbol{v} - \boldsymbol{w}_t) + \boldsymbol{g}_t^T(\boldsymbol{v} - \boldsymbol{w}_t) \tag{9}$$

In the reduced space, the potential new iterate (9) can be represented by,

$$\boldsymbol{z}^+ = \operatorname*{argmin}_{\boldsymbol{x}} h(\boldsymbol{x}) + \frac{1}{2}(\boldsymbol{x} - \boldsymbol{z}_t)^T \hat{B}_t(\boldsymbol{x} - \boldsymbol{z}_t) + \hat{\boldsymbol{g}}_t^T(\boldsymbol{x} - \boldsymbol{z}_t) \tag{10}$$

$\boldsymbol{z}^+$ and $\boldsymbol{w}^+$ also satisfy Equation (5), i.e.

$$\boldsymbol{w}^+ = \operatorname*{argmin}_{U^T \boldsymbol{w} = \boldsymbol{z}^+} \|\boldsymbol{w}\|_1 \tag{11}$$

In this paper, we consider the convergence phase when $\boldsymbol{z}_t$ is close enough to the optimum such that the unit step size is always chosen, i.e. $\boldsymbol{z}_{t+1} = \boldsymbol{z}^+$ [4].

## 1.2 Global linear Convergence

**Lemma 1** (Global linear Convergence). *For $\nabla\hat{\ell}(z)$ satisfying Lipschitz-continuity with a constant $L_g$ and $B_t$ satisfying CNSC-$\mathcal{T}$, the sequence $\{z_t\}_{t=1}^{\infty}$ produced by Prox-QN method converges at least R-linearly.*

*Proof.* This theorem follows Theorem 2 in [6], where the coordinate block $J_k$ is chosen to be the whole coordinate set. Assumption 2(a) in [6] is satisfied because of Theorem 4 C4 in [6] by assuming $\nabla\hat{\ell}(z)$ is Lipschitz-continuous. Other conditions of Theorem 2 in [6] can be easily justified. $\qquad\square$

## 1.3 Quadratic Convergence of Proximal Newton Method and Dennis-More Criterion

**Lemma 2** (Quadratic Convergence of Prox-Newton (Theorem 3 in [1])). *For $\ell(w)$ satisfying CNSC-$\mathcal{T}$ with Lipschitz-continuous second derivative $H(w) = \nabla^2\ell(w)$, the sequence $\{w_t\}$ produced by proximal Newton Method in the quadratic convergence phase has*

$$\|z_{t+1} - z^*\| \leq \frac{L_H}{2m}\|z_t - z^*\|^2,$$

*where $z^* = U^T w^*$, $z_t = U^T w_t$, $w^*$ is the optimal solution and $L_H$ is the Lipschitz constant for $H(w)$.*

**Lemma 3.** *If $B_0 = U\hat{B}_0 U^T$ satisfies CNSC-$\mathcal{T}$ condition, then $\hat{B}_t$ generated by (8) satisfies the Dennis-More criterion [3], namely,*

$$\lim_{t\to\infty} \frac{\|(\hat{B}_t - \hat{H}^*)(z_{t+1} - z_t)\|}{\|z_{t+1} - z_t\|} = 0,$$

*where $\hat{H}^* = \nabla^2\hat{\ell}(z^*)$ and $z^*$ is the optimal solution of (4).*

*Proof.* We want to show that this proof can follow the proof of Theorem 6.6 in [2]. We will verify that the conditions of Theorem 6.6 in [2] are satisfied here. First, the Lipschitz continuity of $\hat{H}(z)$ is implied by Lipschitz continuity of $H(w)$ :

$$
\begin{aligned}
\|(\hat{H}(z_1) - \hat{H}(z_2))\| &= \|U^T(H(w_1) - H(w_2))U\| \\
&\leq \|H(w_1) - H(w_2)\| \\
&= \|H(Uz_1) - H(Uz_2)\| \\
&\leq L_H\|z_1 - z_2\|
\end{aligned}
$$

where the last inequality is from the Lipschitz continuity of $H(w)$. The second condition, $\sum_{t=0}^{\infty}\|z_t - z^*\| < \infty$, is implied by the global linear convergence(Lemma 1). $\qquad\square$

## 1.4 Asymptotic Superlinear Convergence

**Proof of Theorem 1**

*Proof.* If $B_t$ satisfies CNSC-$\mathcal{T}$ condition, then $\hat{B}_t$ satisfies $m\|z\|^2 \leq z^T\hat{B}_t z \leq M\|z\|^2$ for any $z \in \mathbb{R}^{\hat{d}}$. The Lipschitz-continuous $H$ implies Lipschitz-continuity of $\hat{H}$. Therefore by applying the Prox-QN method in the reduced space, this theorem follows Theorem 3.7 in [4], Lemma 3 and Lemma 2. $\qquad\square$

**Proof of Theorem 2**

*Proof.* We prove this theorem by showing $|\ell(w_t) - \ell(w^*)| \leq L_\ell\|z_t - z^*\|$ and $\|w_t\|_1 - \|w^*\|_1 \leq \sqrt{d}\|z_t - z^*\|$. The first part is given by,

$$|\ell(w_t) - \ell(w^*)| = |\ell(UU^T w_t) - \ell(UU^T w^*)| \leq L_\ell\|UU^T(w_t - w^*)\| = L_\ell\|z_t - z^*\|$$

where the inequality comes from the Lipschitz-continuity of $\ell(w)$. In the super-linear convergence phase, the unit step size is chosen, so each iterate satisfies (11). We have $\|w_t\|_1 \leq \|UU^T w_t +$

$(I - UU^T)\boldsymbol{w}^*\|_1$. Moreover, due to the Lipschitz-continuity of $\ell_1$ norm, which is $\|\boldsymbol{w}\|_1 - \|\boldsymbol{v}\|_1 \leq \sqrt{d}\|\boldsymbol{w} - \boldsymbol{v}\|$, we have,

$$\|UU^T\boldsymbol{w}_t + (I - UU^T)\boldsymbol{w}^*\|_1 \leq \|\boldsymbol{w}^*\|_1 + \sqrt{d}\|UU^T\boldsymbol{w}_t - UU^T\boldsymbol{w}^*\|$$

$$\leq \|\boldsymbol{w}^*\|_1 + \sqrt{d}\|\boldsymbol{z}_t - \boldsymbol{z}^*\|$$

$\square$

## 1.5 Global Convergence with Shrinking

In Theorem 1, we assume shrinking strategy is not employed and the inner problem is solved exactly. In this subsection, we show that by only assuming the inner problem is solved exactly, Prox-QN method with shrinking will still globally converge to the optimum under the CNSC-$\mathcal{T}$ condition. We first prove that with sufficient small step size, the Armijo rule will be satisfied.

**Lemma 4.** *If the step size,*

$$\alpha \leq \min\{1, \frac{m}{L_1}(1 - \sigma)\}$$

*then the Armijo rule is satisfied, i.e.,*

$$f(\boldsymbol{w} + \alpha\boldsymbol{d}) \leq f(\boldsymbol{w}) + \alpha\sigma(\lambda\|\boldsymbol{w} + \boldsymbol{d}\|_1 - \lambda\|\boldsymbol{w}\|_1 + \boldsymbol{g}^T\boldsymbol{d})$$

*where $L_1$ is the Lipschitz-continuity constant.*

*Proof.* Let $\boldsymbol{w}^+ = \boldsymbol{w} + \alpha\boldsymbol{d}$,

$$f(\boldsymbol{w}^+) - f(\boldsymbol{w}) = \ell(\boldsymbol{w}^+) - \ell(\boldsymbol{w}) + \lambda(\|\boldsymbol{w}^+\|_1 - \|\boldsymbol{w}\|_1)$$

$$\leq \int_0^1 \nabla\ell(\boldsymbol{w} + s\alpha\boldsymbol{d})(\alpha\boldsymbol{d})ds + \alpha\lambda\|\boldsymbol{w} + \boldsymbol{d}\|_1 + (1 - \alpha)\lambda\|\boldsymbol{w}\|_1 - \lambda\|\boldsymbol{w}\|_1$$

$$= \alpha(\nabla\ell(\boldsymbol{w})^T\boldsymbol{d} + \lambda\|\boldsymbol{w} + \boldsymbol{d}\|_1 - \lambda\|\boldsymbol{w}\|_1) + \alpha\int_0^1 \boldsymbol{d}^T(\nabla\ell(\boldsymbol{w} + s\alpha\boldsymbol{d}) - \nabla\ell(\boldsymbol{w}))ds$$

$$\leq \alpha(\nabla\ell(\boldsymbol{w})^T\boldsymbol{d} + \lambda\|\boldsymbol{w} + \boldsymbol{d}\|_1 - \lambda\|\boldsymbol{w}\|_1) + \alpha\int_0^1 \|U^T\boldsymbol{d}\|\|\nabla\ell(\boldsymbol{w} + s\alpha\boldsymbol{d}) - \nabla\ell(\boldsymbol{w})\|ds$$

Because

$$\|\nabla\ell(\boldsymbol{w} + s\alpha\boldsymbol{d}) - \nabla\ell(\boldsymbol{w})\| = \|\nabla\ell(UU^T\boldsymbol{w} + s\alpha UU^T\boldsymbol{d}) - \nabla\ell(UU^T\boldsymbol{w})\| \leq sL_1\|U^T\boldsymbol{d}\|$$

we have

$$f(\boldsymbol{w}^+) - f(\boldsymbol{w}) \leq \alpha\left((\nabla\ell(\boldsymbol{w})^T\boldsymbol{d} + \lambda\|\boldsymbol{w} + \boldsymbol{d}\|_1 - \lambda\|\boldsymbol{w}\|_1) + \frac{L_1\alpha}{2}\|U^T\boldsymbol{d}\|^2\right)$$

For $\alpha \leq \min\{1, \frac{m}{L_1}(1 - \sigma)\}$,

$$\frac{L_1\alpha}{2}\|U^T\boldsymbol{d}\|^2 \leq \frac{m}{2}(1 - \sigma)\|U^T\boldsymbol{d}\|^2 \leq \frac{1 - \sigma}{2}\boldsymbol{d}^T B\boldsymbol{d}$$

As $\boldsymbol{d}$ minimizes Eq. (2) in the main paper, we have $\frac{1}{2}\boldsymbol{d}^T B\boldsymbol{d} \leq -(\nabla\ell(\boldsymbol{w})^T\boldsymbol{d} + \lambda\|\boldsymbol{w} + \boldsymbol{d}\|_1 - \lambda\|\boldsymbol{w}\|_1)$. So we obtain the sufficient descent condition,

$$f(\boldsymbol{w}^+) - f(\boldsymbol{w}) \leq \alpha\sigma\left(\nabla\ell(\boldsymbol{w})^T\boldsymbol{d} + \lambda\|\boldsymbol{w} + \boldsymbol{d}\|_1 - \lambda\|\boldsymbol{w}\|_1\right)$$

$\square$

**Proposition 1.** *Assume $\nabla^2\ell(\boldsymbol{w})$ and $\nabla\ell(\boldsymbol{w})$ are Lipschitz continuous. Let $\{B_t\}_{t=1,2,3...}$ be the matrices generated by BFGS update. Then if $\ell(\boldsymbol{w})$ and $B_t$ satisfy CNSC-$\mathcal{T}$ condition and the inner problem is solved exactly, the proximal quasi-Newton method with shrinking has global convergence.*

*Proof.* Our algorithm allows all the variables to re-enter the working set at the beginning of each epoch. And before it terminates all the variables must be checked. Thus as many as epochs are taken in the optimization procedure until the global stopping criterion is attained. Let's denote $\{t_k\}_{k=0,1,2,3...}$ to be the iterations when an epochs begins. In these iterations, all the variables are taken into consideration. As shown in Lemma 4, there exists some constant $\alpha_0$,

$$f(\boldsymbol{w}_{t_k+1}) - f(\boldsymbol{w}_{t_k}) \leq \alpha_0 \sigma \left( \nabla\ell(\boldsymbol{w}_{t_k})^T \boldsymbol{d}_{t_k} + \lambda\|\boldsymbol{w}_{t_k} + \boldsymbol{d}_{t_k}\|_1 - \lambda\|\boldsymbol{w}_{t_k}\|_1 \right)$$

And as in each epoch the function value is non-increasing across the iterations, i.e. for any $k$, $f(\boldsymbol{w}_{t_{k+1}}) \leq f(\boldsymbol{w}_{t_k+1})$. Thus, we have

$$f(\boldsymbol{w}_{t_K+1}) - f(\boldsymbol{w}_{t_0}) \leq \sum_{k=0}^{K} f(\boldsymbol{w}_{t_k+1}) - f(\boldsymbol{w}_{t_k}) \leq -\alpha_0\sigma \sum_{k=0}^{K} \boldsymbol{d}_{t_k}^T B_{t_k} \boldsymbol{d}_{t_k}$$

As $f(\boldsymbol{w}_{t_K+1}) - f(\boldsymbol{w}_{t_0}) > -\infty$, $\lim_{k\to\infty} \boldsymbol{d}_{t_k}^T B_{t_k} \boldsymbol{d}_{t_k} = 0$. Thus, $U^T \boldsymbol{d}_{t_k} \to \mathbf{0}$. That is to say, $\lim_{k\to\infty} \boldsymbol{d}_{t_k} \in \mathcal{T}^\perp$. If $\boldsymbol{d}_t \in \mathcal{T}^\perp$, the line search procedure will always pick unit step size. And in the next iteration, $\boldsymbol{d}_{t+1} = 0$. So when $U^T \boldsymbol{d}_{t_k} \to \mathbf{0}$, we also have $\boldsymbol{d}_{t_k} \to \mathbf{0}$. Therefore, $\boldsymbol{w}_{t_k}$ converges to the optimum according to Proposition 2.5 in [4].

$\square$

## 2 Algorithm Details

---

**Algorithm 1** Proximal Quasi-Newton Algorithm

---

**Input:** Observations $\{\boldsymbol{x}^{(i)}\}_{i=1,2,...,N}$, labels $\{\boldsymbol{y}^{(i)}\}_{i=1,2,...,N}$, termination criterion $\epsilon$, scalar $\lambda$ and L-BFGS memory size $m$.

**Output:** $\boldsymbol{w}^*$ converging to arg $\min_{\boldsymbol{w}} f(\boldsymbol{w})$

1: Initialize $\gamma = 1$, $\boldsymbol{w} \leftarrow \boldsymbol{0}$, $\boldsymbol{g} \leftarrow \partial\ell(\boldsymbol{w})/\partial\boldsymbol{w}$, working set $\mathcal{A} \leftarrow \{1, 2, ...d\}$, $\hat{M} \leftarrow \infty$, and $S$, $Y$, $Q$, $\hat{Q} \leftarrow \phi$.

2: **for** $n = 0, 1, ...$ **do**

3:     $\hat{\mathcal{A}} \leftarrow \mathcal{A}$, $\mathcal{A} \leftarrow \phi$, $M \leftarrow 0$

4:     **for** $j$ in $\hat{\mathcal{A}}$ **do**                    ▷ Shrink the working set

5:         calculate $\partial_j f$

$$\partial_j f(\boldsymbol{w}) = \begin{cases} g_j + \text{sgn}(w_j)\lambda & \text{if } w_j \neq 0 \\ \text{sgn}(g_j) \max\{|g_j| - \lambda, 0\} & \text{if } w_j = 0 \end{cases} \tag{12}$$

6:         **if** $w_j \neq 0$ or $|g_j| - \lambda + \hat{M}/N > 0$ **then**

7:             $\mathcal{A} \leftarrow \mathcal{A} \cup j$, $M \leftarrow \max\{M, |\partial_j f|\}$

8:         **end if**

9:     **end for**

10:     $\hat{M} \leftarrow M$

11:     **if** Shrinking stopping criterion attained **then**      ▷ Check shrinking stopping criterion

12:         **if** Stopping criterion attained and $|\hat{\mathcal{A}}| = d$ **then**      ▷ Check global stopping criterion

13:             return $\boldsymbol{w}$

14:         **else**

15:             $\boldsymbol{g} \leftarrow \partial\ell(\boldsymbol{w})/\partial\boldsymbol{w}$, $\mathcal{A} \leftarrow \{1, 2, ...d\}$ and $S$, $Y$, $Q$, $\hat{Q} \leftarrow \phi$

16:             Update shrinking stopping criterion and then continue

17:         **end if**

18:     **end if**

19:     $\boldsymbol{d} \leftarrow \boldsymbol{0}$, $\hat{\boldsymbol{d}} \leftarrow \boldsymbol{0}$

20:     Compute $inner\_iter = \min\{max\_inner, \lfloor\frac{d}{|\mathcal{A}|}\rfloor\}$

21:     **for** $p = 1, 2, ...inner\_iter$ **do**              ▷ Solve inner problem

22:         **for** $j$ in $\mathcal{A}$ **do**

23:             $B_{jj} = \gamma - \boldsymbol{q}_j^T \hat{\boldsymbol{q}}_j$, $(Bd)_j = \gamma d_j - \boldsymbol{q}_j^T \hat{\boldsymbol{d}}$

24:             $a = (B_t)_{jj}$, $b = (\boldsymbol{g}_t)_j + (B_t\boldsymbol{d})_j$ and $c = (\boldsymbol{w}_t)_j + d_j$

25:             Compute $z$ according to $z = -c + \mathcal{S}(c - b/a, \lambda/a)$

26:             $d_j \leftarrow d_j + z$, $\hat{\boldsymbol{d}} \leftarrow \hat{\boldsymbol{d}} + z\hat{\boldsymbol{q}}_j$

27:         **end for**

28:     **end for**

29:     **for** $\alpha = \beta^0, \beta^1, ....$ **do**              ▷ Conduct line search

30:         **if** $f(\boldsymbol{w} + \alpha\boldsymbol{d}) \leq f(\boldsymbol{w}) + \alpha\sigma(\lambda\|\boldsymbol{w} + \boldsymbol{d}\|_1 - \lambda\|\boldsymbol{w}\|_1 + \boldsymbol{g}^T\boldsymbol{d})$ **then**

31:             break

32:         **end if**

33:     **end for**

34:     **for** $j$ in $\mathcal{A}$ **do**

35:         $g_j^{new} = \partial\ell(\boldsymbol{w})/\partial w_j$, $y_j = g_j^{new} - g_j$, $s_j = \alpha d_j$, $g_j = g_j^{new}$

36:     **end for**

37:     Update $S$, $Y$ and $Q$ just on the rows corresponding to $\mathcal{A}$.

38:     Update $\gamma$, $D$, $L$, $S^T S$ where the inner product between **s** and another vector is computed just over $\mathcal{A}$.

39:     Update $R$ and then update $\hat{Q}$ just on the columns corresponding to $\mathcal{A}$.

40: **end for**

---

# 3 Proof of Theorem 3

*Proof.* The Hessian of $\ell(\boldsymbol{w})$ for CRF MLEs is

$$H = \sum_{i=1}^{N} \left( E\left[\phi(\boldsymbol{y}, \boldsymbol{x}^{(i)})\phi(\boldsymbol{y}, \boldsymbol{x}^{(i)})^T\right] - E\left[\phi(\boldsymbol{y}, \boldsymbol{x}^{(i)})\right] E\left[\phi(\boldsymbol{y}, \boldsymbol{x}^{(i)})\right]^T \right), \qquad (13)$$

where $\phi(\boldsymbol{y}, \boldsymbol{x}^{(i)}) = \left[f_1(\boldsymbol{y}, \boldsymbol{x}^{(i)}), f_2(\boldsymbol{y}, \boldsymbol{x}^{(i)}), ..., f_d(\boldsymbol{y}, \boldsymbol{x}^{(i)})\right]^T$ and $E$ is the expectation over the conditional probability $P_{\boldsymbol{w}}(\boldsymbol{y}|\boldsymbol{x}^{(i)})$. Now we re-formulate (13) to

$$H = \Phi D \Phi$$

Here $D \in \mathbb{R}^{(N|\mathcal{Y}|) \times (N|\mathcal{Y}|)}$ is a diagonal matrix with diagonal elements $D_{nn} = P_{\boldsymbol{w}}(\boldsymbol{y}_l|\boldsymbol{x}^{(i)})$, where $n = (i-1)|\mathcal{Y}| + l$ and $l = 1, 2, .., |\mathcal{Y}|$. $\Phi$ is a $d \times (N|\mathcal{Y}|)$ matrix whose column $n$ is defined as $\Phi_n = \phi(\boldsymbol{y}_l, \boldsymbol{x}^{(i)}) - E\left[\phi(\boldsymbol{y}, \boldsymbol{x}^{(i)}\right]$ for $n = (i-1)|\mathcal{Y}| + l$.

The theorem holds because of the following four reasons.
**a. $\mathcal{N}$ is constant with respect to $\boldsymbol{w}$.**
$\mathcal{N}$ is equivalent to

$$\mathcal{N} = \{\boldsymbol{a} \in \mathbb{R}^d | \forall i, \exists \text{ some constant } b_i, \langle \boldsymbol{a}, \phi(\boldsymbol{y}, \boldsymbol{x}^{(i)})\rangle = b_i \text{ for } \forall \boldsymbol{y}\} \qquad (14)$$

Thus $\mathcal{N}$ is independent on $\boldsymbol{w}$ and so is $\mathcal{T}$.
**b. $\ell(\boldsymbol{w})$ depends only on $\boldsymbol{z} = \text{proj}_{\mathcal{T}}(\boldsymbol{w})$.**
Let $\boldsymbol{w} = \boldsymbol{z} + \boldsymbol{u}$. So $\boldsymbol{u} \in \mathcal{N}$.

$$\begin{aligned}
P_{\boldsymbol{w}}(\boldsymbol{y}^{(i)}|\boldsymbol{x}^{(i)}) &= \frac{\exp\left\{\langle \boldsymbol{w}, \phi(\boldsymbol{y}^{(i)}, \boldsymbol{x}^{(i)})\rangle\right\}}{\sum_{\boldsymbol{y}} \exp\left\{\langle \boldsymbol{w}, \phi(\boldsymbol{y}, \boldsymbol{x}^{(i)})\rangle\right\}} \\
&= \frac{\exp\left\{\langle \boldsymbol{z}, \phi(\boldsymbol{y}^{(i)}, \boldsymbol{x}^{(i)})\rangle\right\} \exp\left\{\langle \boldsymbol{u}, \phi(\boldsymbol{y}^{(i)}, \boldsymbol{x}^{(i)})\rangle\right\}}{\sum_{\boldsymbol{y}} \exp\left\{\langle \boldsymbol{z}, \phi(\boldsymbol{y}, \boldsymbol{x}^{(i)})\rangle\right\} \exp\left\{\langle \boldsymbol{u}, \phi(\boldsymbol{y}, \boldsymbol{x}^{(i)})\rangle\right\}} \\
&= \frac{\exp\left\{\langle \boldsymbol{z}, \phi(\boldsymbol{y}^{(i)}, \boldsymbol{x}^{(i)})\rangle\right\}}{\sum_{\boldsymbol{y}} \exp\left\{\langle \boldsymbol{z}, \phi(\boldsymbol{y}, \boldsymbol{x}^{(i)})\rangle\right\}}
\end{aligned}$$

The last equality comes from the character of $\mathcal{N}$, Equation (14).

**c. The first property Eq. (2) holds.**
$D_{nn} \to 0$ iff $\|\boldsymbol{w}\|_1 \to \infty$ which is prohibited by $\ell_1$ penalty. Thus there exists $m_p > 0$ such that $D_{nn} \geq m_p$ for any $n$. Hence, the positive definiteness of $H$ is determined by $\Phi$.
So we have for any $\boldsymbol{v} \in \mathcal{T}$,

$$m_p \lambda_{min}(\Phi\Phi^T)\|\boldsymbol{v}\|^2 \leq m_p \boldsymbol{v}^T \Phi\Phi^T \boldsymbol{v} \leq \boldsymbol{v}^T H \boldsymbol{v} \leq \boldsymbol{v}^T \Phi\Phi^T \boldsymbol{v} \leq \lambda_{max}(\Phi\Phi^T)\|\boldsymbol{v}\|^2$$

where $\lambda_{min}(\Phi\Phi^T)$ is the minimum nonzero eigenvalue of $\Phi\Phi^T$ and $\lambda_{max}(\Phi\Phi^T)$ is the maximum eigenvalue of $\Phi\Phi^T$.
**d. The second property Eq. (3) holds.**
This property directly follows the definition of $\mathcal{N}$.

$\square$

# 4 Gradient evaluation in sequence labeling and hierarchical classification

The gradients for general CRF problems are given by

$$\frac{\partial \ell(\boldsymbol{w})}{\partial w_k} = \sum_{i=1}^{N} \left( \sum_{\boldsymbol{y} \in \mathcal{Y}} P_{\boldsymbol{w}}(\boldsymbol{y}|\boldsymbol{x}^{(i)}) f_k(\boldsymbol{y}, \boldsymbol{x}^{(i)}) - f_k(\boldsymbol{y}^{(i)}, \boldsymbol{x}^{(i)}) \right) \qquad (15)$$

### 4.1 Sequence labeling

The partial gradients of $\ell(\boldsymbol{w})$ for sequence labeling problem are,

$$\frac{\partial l(\Theta, \Lambda)}{\partial \Theta_{y,j}} = \sum_{i=1}^{N} \sum_{t=1}^{T^{(i)}} \left( P_{\boldsymbol{w}}(y_t = y | \boldsymbol{x}^{(i)}) - \mathbf{1}\left[y_t^{(i)} = y\right] \right) x_{tj}^{(i)} \tag{16}$$

$$\frac{\partial l(\Theta, \Lambda)}{\partial \Lambda_{y,y'}} = \sum_{i=1}^{N} \sum_{t=1}^{T^{(i)}-1} \left( P_{\boldsymbol{w}}(y_t = y, y_{t+1} = y' | \boldsymbol{x}^{(i)}) - \mathbf{1}\left[y_t^{(i)} = y, y_{t+1}^{(i)} = y'\right] \right) \tag{17}$$

The forward-backward algorithm is a popular inference oracle for evaluating the marginal probability in Equation (16) and (17). In our OCR model, the forward-backward algorithm is

$$\begin{cases} \alpha_1(y) = \exp(\Theta_y^T \boldsymbol{x}_1) \\ \alpha_{t+1}(y) = \sum_{y'} \alpha_t(y') \exp(\Theta_y^T \boldsymbol{x}_{t+1} + \Lambda_{y',y}) \end{cases}$$

$$\begin{cases} \beta_T(y) = 1 \\ \beta_t(y') = \sum_{y} \beta_{t+1}(y) \exp(\Theta_y^T \boldsymbol{x}_{t+1} + \Lambda_{y',y}) \end{cases}$$

where $\Theta_y^T$ is the y-th row of the matrix $\Theta$. Then the marginal conditional probabilities are given by

$$P_{\boldsymbol{w}}(y_t = y', y_{t+1} = y | \boldsymbol{x}) = \frac{1}{Z_{\boldsymbol{w}}(\boldsymbol{x})} \alpha_t(y') \exp(\Theta_y \boldsymbol{x}_{t+1} + \Lambda_{y',y}) \beta_{t+1}(y)$$

$$P_{\boldsymbol{w}}(y_t = y | \boldsymbol{x}) = \frac{1}{Z_{\boldsymbol{w}}(\boldsymbol{x})} \alpha_t(y) \beta_t(y),$$

where the normalization factor $Z_{\boldsymbol{w}}(\boldsymbol{x})$ can be computed by $\sum_y \alpha_T(y)$.

### 4.2 Hierarchical classification

The partial gradients of $\ell(W)$ for hierarchical classification problem are,

$$\frac{\partial \ell(W)}{\partial W_{k,j}} = \sum_{i=1}^{N} \left( \sum_{y \in \mathcal{Y}} \mathbf{1}\left[k \in \text{Path}(y)\right] P_W(y | \boldsymbol{x}^{(i)}) - \mathbf{1}\left[k \in \text{Path}(y^{(i)})\right] \right) x_j^{(i)}$$

They can be evaluated by the downward-upward algorithm. Let $\alpha(k)$ and $\beta(k)$ be the downward message and upward message respectively.

$$\begin{cases} \alpha(root) = \boldsymbol{w}_{root}^T \boldsymbol{x} \\ \alpha(k) = \alpha(\text{parent}(k)) + \boldsymbol{w}_k^T \boldsymbol{x} \end{cases}$$

$$\begin{cases} \beta(k) = \alpha(k) / \sum_{y \in Y} \alpha(y) & \text{if } k \text{ is a leaf node} \\ \beta(k) = \sum_{k' \in \text{children}(k)} \beta(k') & \text{if } k \text{ is a non-leaf node} \end{cases}$$

So we have

$$\frac{\partial \ell(W)}{\partial W_{k,j}} = \sum_{i=1}^{N} \left( \beta^{(i)}(k) - \mathbf{1}\left[k \in \text{Path}(y^{(i)})\right] \right) x_j^{(i)} \tag{18}$$

## 5 More Experimental Results

### 5.1 Performance on different values of $\lambda$

$\lambda$ affects the sparsity of the intermediate iterates, and further the speed of the algorithm. In particular, when $\lambda$ is larger, the intermediate iterates are sparser, and then the corresponding iterations, due to the shrinking strategy, will be faster – and vice versa. The effect of $\lambda$ on the performance is shown in this section.

### 5.1.1 Sequence Labeling

(a) Relative Objective Difference

(b) Non-zero Parameters

Figure 1: Sequence Labeling Problem for $\lambda = 500$

(a) Relative Objective Difference

(b) Non-zero Parameters

Figure 2: Sequence Labeling Problem for $\lambda = 50$

### 5.1.2 Hierarchical Classification

(a) Relative Objective Difference      (b) Non-zero Parameters

Figure 3: Hierarchical Classification Problem for $\lambda = 2$

(a) Relative Objective Difference      (b) Non-zero Parameters

Figure 4: Hierarchical Classification Problem for $\lambda = 0.5$

### 5.2 Testing Accuracy

The testing accuracy for different $\lambda$'s on these two problems is in the following tables. The testing accuracy across training time is shown in Figure 5.

| $\lambda$ | 50 | 100 | 500 |
|---|---|---|---|
| Testing Accuracy | 0.834928 | 0.736643 | 0.407895 |
| nnz of optimum | 2542 | 1544 | 223 |

Table 1: Per-character testing accuracy for OCR dataset

| $\lambda$ | 0.5 | 1 | 2 |
|---|---|---|---|
| Testing Accuracy | 0.262648 | 0.249731 | 0.185684 |
| nnz of optimum | 28483 | 4301 | 1505 |

Table 2: Testing accuracy for LSHTC1 dataset

(a) OCR dataset with $\lambda = 100$       (b) LSHTC1 dataset with $\lambda = 1$

Figure 5: Testing accuracy v.s. training time

## 5.3 Relative objective difference v.s. the number of passes over dataset

Figure 6 shows the performance under the measure of number of passes (iterations) over dataset. We do not include the plot for BCD method, because the pass over dataset for BCD actually depends on the sparsity pattern of the dataset. Thus it is hard to fairly define the pass over dataset for BCD.

Figure 6: Relative objective difference v.s. the number of passes over dataset for OCR dataset with $\lambda = 100$.