[Reviews · NeurIPS 2014]

Submitted by Assigned_Reviewer_8

This paper describes a new proximal quasi-Newton algorithm for optimizing L1-regularized M-estimators (e.g., application include L1-regularized CRFs). The basic approach goes beyond standard quasi-Newton optimization algorithms by (1) including an L1-proximal term for stabilizing the search directions for the optimization algorithm, (2) solving for the search direction using randomized coordinate descent, (3) employing an aggressive shrinking strategy for keeping the active set small at most times, and (4) using data structures that exploit both sparsity in the data features and the learned model. The authors also describe conditions under which the algorithm is super-linearly convergent.

The method described is elegant, but even more impressive is the performance of the technique on actual data. In two real-world examples, the proposed algorithm seems to perform substantially better than existing approaches. However, it’s unclear to me how the regularization parameters were set for the experiments; in particular, were they chosen to optimize generalization performance, or to illustrate the dependence of other algorithms on strong convexity for reasonable performance? Separately, do the stark differences in performance remain if one considers not optimization objective but generalization performance on test data?

Nonetheless, I found the method described in this paper to be very interesting and insightful, regardless of the empirical testing.
Summary: An interesting twist on Quasi-Newton algorithms for L1-regularized problems that exploits sparsity well to achieve super-linear convergence. Compelling experiments.

Submitted by Assigned_Reviewer_16

This paper builds on the idea of constant nullspace strong convexity (CNSC) which is proposed in (seemingly) a parallel submission. The idea of this is quite natural-- given a dataset of size, say, 10, a linear classifier might be strongly convex in the span of the 10 inputs, but have no dependence outside this space. A function is said to have CNSC if there is a subspace like this over which it is strongly convex, but independent outside the space.

This particular paper looks at applying CNSC to a quasi-newton (BFGS) algorithm with an l_1 regularizer. The main theoretical result is theorem 1, which says that, asymptotically, convergence will be superlinear. There are also quite strong experimental results.

A strong aspect of this paper is the mix of theory with very precise descrptions of implementation details that (seemingly) are important for good empirical performance.

CNSC seems an exciting idea, and, to the best of my knowledge, this is the first proof that a quasi-newton method will convergence superllinearly in situations where it applys. It is slightly difficult to determine the exact contribution of this paper without access to the first (anonymous) reference, but there doesn't seem to be too much overlap.

As a minor comment, it would also be useful to see separate plots of the convergence rates of the different optimization methods as a function of the # of iterations, and the amount of time per iteration (also as a function of the # of iterations). This allows a reader to disambiguate improvements in computational speed from those in optimization convergence rates.

As a minor comment, there are a few errors in the typesetting, typically involved quantities with underscores.

EDIT: I'd also like to echo the point that the experiments would probably be better measuring "number of passes over the dataset" rather than time. I realize, of course, that time is what one is actually interested in, but passes over data is a measure that generalises much better to different implementations/problems/etc.
Summary: A strong paper, I vote for acceptance

Submitted by Assigned_Reviewer_23

## Summary

The paper introduces a proximal quasi Newton algorithm for the solution of
computationally intensive l1-regularized m-estimators. CRFs are an example of
such a setting (it is relatively slow to compute the function value and/or
gradient of an instance). The main contribution is a superlinear convergence
proof of the algorithm under the assumption of strong convexity in a subspace of
the original optimization.

## Specific Points

- it's Lipschitz, not Lipchitz!
- 134: one-dimensional problem
- 183: of the inner problem
- 201: include a pointer to the appendix in the caption?
- 241: q-superlinear without q appearing in the formula...
- 249: state assumptions onn \ell(w) again?
- 259: CRF problems
- 321: one-dimensional
- 330: only in very few blocks
- 354: I'd be instered to see how performance is for different \lambda and
possibly also different feature functions (say non-quadratic unary features)
to get an understanding of whether the algorithm improves performance across
the board.
- 374: why no cross-validation? Or at least show results for several \lambda
- 408: how do you choose \lambda=2 and \eta_0=0.1? Does performance change if
you alter \lambda?
- 427: I feel it is rather unfair to include the time for the whole function
evaluation in SGD, can't you just *not* include the time required to compute the
objective for plotting purposes? Or show plots with epochs, rather than time
on the x-axis.
- 429: rephrase second part of sentence
- [430: short discussion/summary is always appreciated]

## Quality

The experiments should be improved (fix SGD and include some other settings of
\lambda). Also, it is not obvious to me whether the convergence rate still hold
for the actual implementation of Prox-QN with the inexact inner problem solution /
shrinking strategy.

## Clarity

The paper is well written and clear in most parts. I would appreciate some more
details in the main paper about the shrinking strategy. It is also not clear to
me how the shrinking strategy comes into play with the convergence proof?

## Originality

As far as I can say the work is novel. Overlap with [1] might be something to
worry about, because there might the CNSC part is probably present in both
works.

## Significance

The experiments seem to support the superlinear convergence, which is quite
impressive (a plot with test error might though still be interesting). Efficient
training of l1-regularized CRFs is an important problem and such substantial
speedups make the work significant.
Summary: While it seems the papers contributions are mostly in combining/extending
results, the convergence rate in the experiments/theory is quite impressive and
justifies acceptance.
Author Feedback
Author rebuttal: We are grateful to the reviewers for their careful and constructive comments.

1. Assigned_Reviewer_16 and Assigned_Reviewer_23: regarding possible overlap with the anonymous reference [1].

[1] analyzes the behavior of proximal gradient and Newton methods under CNSC, while this paper analyzes the trickier proximal quasi-Newton, whose asymptotic super-linear convergence proof is quite different from that for proximal gradient and Newton methods.

2. Assigned_Reviewer_23 and Assigned_Reviewer_8: regarding how the regularization parameter \lambda is set and how performance varies for different \lambda.

\lambda affects the sparsity of the intermediate iterates, and further the speed of the algorithm.
In particular, when \lambda is larger, the intermediate iterates are sparser, and then the corresponding iterations, due to the shrinking strategy, will be faster -- and vice versa.
The \lambda's in this paper are chosen to be not too large such that the number of nonzeros of the learned model is reasonable, and at the same time, not too small such that the sparsity of most intermediate iterates is still maintained.

When \lambda is very small, the convergence of our method is slow at the beginning but becomes fast when the iterate is close to the optimal.
For example, here is the time (seconds) used by different methods to achieve different relative objective difference when \lambda = 1 (\lambda=500 in the paper) for the sequence labeling.

Relative objective difference | Prox-QN | OWL-QN | BCD | SGD |
1e-1 | 939 | 1317 | 9559| * |
1e-2 | 1218 | 6424 | * | * |
1e-3 | 1459 | * | * | * |
* denotes the time is greater than 100,000s.

In this case, the learning rate, \eta_0, for SGD is tuned to be 3e-2 for optimal speed performance. The number of nonzeros in the optimal solution is 12,278 (276 when \lambda=500).

We will provide empirical results for different values of \lambda in the final version.

Assigned_Reviewer_23 also asked how \eta_0=0.1 is set in the hierarchical classification experiment. As in the sequence labeling experiment, it is tuned for optimal speed performance.

3. Assigned_Reviewer_23: "It is also not clear to me how the shrinking strategy comes into play with the convergence proof?" "Also, it is not obvious to me whether the convergence rate still hold for the actual implementation of Prox-QN with the inexact inner problem solution / shrinking strategy."

In the theorems, we assume the inner problem is solved exactly and shrinking strategy is not applied. We will highlight these assumptions in the theorems. Due to the use of epochs in the shrinking strategy which allow all the shrunk variables to re-enter, the convergence analysis should extend to the case with the shrinking strategy, but would make the presentation more complex. We will include the result in the paper, and add the more complicated presentation in the appendix. Incorporating inexact solutions of inner problem would be much more complex.

4. Assigned_Reviewer_23: "I feel it is rather unfair to include the time for the whole function evaluation in SGD, can't you just *not* include the time required to compute the objective for plotting purposes?"

It is indeed unfair for SGD if the whole function value is calculated every iteration. Thus, we calculated it every 5N iterations, i.e. every 5 epochs. The objective evaluation time is much less than that of 5 epochs. For example, in the sequence labeling experiment, the evaluation time for the whole function value is 1.1 seconds, while the time of 5 epochs is 61 seconds. However, the reviewer's suggestion is better and we will do that in the final version.

5. Assigned_Reviewer_16: "I'd also like to echo the point that the experiments would probably be better measuring "number of passes over the dataset" rather than time."

The number of passes over the dataset is indeed a good measure if one wants to see the super-linear convergence of Prox-QN or to compare the convergence rates of different methods.
However, it is probably unfair for some methods if the number of passes is used as a measure because the time complexities of one pass of different methods are quite different. We will add these plots to the supplementary section in the final version of the paper.

6. Assigned_Reviewer_23: "a plot with test error might though still be interesting". Assigned_Reviewer_8: "Separately, do the stark differences in performance remain if one considers not optimization objective but generalization performance on test data?"

We agree that performance on test data, like test errors, could be interesting as well. We will add these plots to the supplementary section in the final version of the paper.

Again, we appreciate the helpful comments and suggestions by all the reviewers.